# *Phoenix dactylifera* L. Seed Extract Exhibits Antioxidant Effects and Attenuates Melanogenesis in B16F10 Murine Melanoma Cells by Downregulating PKA Signaling

**DOI:** 10.3390/antiox9121270

**Published:** 2020-12-14

**Authors:** Huey-Chun Huang, Shr-Shiuan Wang, Tsang-Chi Tsai, Wang-Ping Ko, Tsong-Min Chang

**Affiliations:** 1Department of Medical Laboratory Science and Biotechnology, China Medical University, Taichung City 406040, Taiwan; lchuang@mail.cmu.edu.tw; 2Department of Applied Cosmetology, HungKuang University, Taichung City 43302, Taiwan; bear9038864@hk.edu.tw; 3O’right Plant Extract R&D Center, Hair O’right International Corporation, Taoyuan City 32544, Taiwan; tsangchi_tsai@oright.com.tw (T.-C.T.); Steven_Ko@oright.com.tw (W.-P.K.)

**Keywords:** *Phoenix dactylifera*, tyrosinase, melanin, ROS, PKA

## Abstract

Background: The mode of action of *Phoenix dactylifera* seed extract in skin care has never been explored. Methods: *P. dactylifera* L. seeds were extracted by ultrasonic extraction. The antioxidant characteristics of the extract were determined by 2,2-diphenyl-1-picrylhydrazyl (DPPH) and 2,2′-azino-di-(3-ethylbenzthiazoline sulfonic acid) (ABTS^+^) assays and scavenging methods. The total phenolic content, reducing capacity, iron (II) ion-chelation, and intracellular reactive oxygen species (ROS)-scavenging capacities were also investigated. The effects of *P. dactylifera* L. seed extract on melanogenesis were evaluated spectrophotometrically by a mushroom tyrosinase activity assay, determination of intracellular tyrosinase activity, and melanin content. The expression levels of melanogenesis-related proteins were analyzed by Western blotting. Results: The results revealed that the *P. dactylifera* L. seed extract exerted apparent antioxidant capacity and significantly decreased intracellular ROS content at concentrations of 0.245 and 0.49 (mg/mL). Furthermore, the extract decreased the expression of melanocortin 1 receptor (MC1R), microphthalmia-associated transcription factor (MITF), tyrosinase, tyrosinase-related protein-1 (TRP1), and tyrosinase-related protein-2 (TRP2), and inhibited melanogenesis in B16F10 cells. Conclusions: Our results revealed that *P. dactylifera* L. seed extract attenuated melanogenesis in B16F10 cells by downregulating protein kinase A (PKA) signaling pathways. Hence, the extract could be used as a type of skin-whitening agent in skin care products.

## 1. Introduction

Antioxidants have been widely applied to prevent or treat oxidative stress-related disorders in cosmetic and dermatological fields. Over the last few decades, antioxidants have also been used in the cosmetic industry to prevent or delay skin aging. It is reported that free radicals and reactive oxygen species (ROS) are associated with several diseases, such as aging and age-related diseases [1]. Interestingly, free radical damage on the skin caused by UV-irradiation stress and ROS plays an important role in the process of skin photoaging [2,3]. Antioxidants are reported to interfere with the oxidation process by scavenging free radicals and ROS or by chelating oxidation-catalytic metals [4]. Hence, there has been a dramatic increase in the number of applications of antioxidants or antioxidant nutritional supplements to reduce oxidative stress or oxidative stress-induced damage in the body [5,6]. However, some chemical antioxidants, including sodium ascorbate, *tert*-butyl hydroxyanisole (BHA), sodium erythorbate, and *tert*-butyl hydroxytoluene (BHT), have been shown to promote carcinogenic effects on human health [7]. Therefore, rapid growth of studies on plant-derived natural antioxidants has occurred over the past decades. Importantly, it was found that increases in the level of ROS can accelerate skin pigmentation. Among the ROS derived from melanocytes and keratinocytes, nitric oxide, NO, stimulates melanin synthesis by enhancing protein expression levels of tyrosinase and tyrosinase-related protein 1 (TRP1) [8,9]. The effect of ROS on melanin production has been studied using various antioxidants, such as N-acetyl cysteine, to abolish the action of UVB-induced α-melanocyte-stimulating hormone (α-MSH) [10].

Melanin is produced and secreted by melanocytes that are distributed in the basal layer of the skin epidermis [11]. The first two steps of the melanogenesis pathway in humans are the hydroxylation of L-tyrosine to 3-4-dihydroxy phenylalanine (L-DOPA) and the oxidation of L-DOPA to o-dopaquinone. Tyrosinase (EC 1.14.18.1) is a rate-limiting enzyme that catalyzes both of the first two reactions during melanogenesis [12]. Melanin plays a vital part in protecting the skin against sunlight ultraviolet (UV) damage and is also responsible for coloring hair, skin, and eyes. It has been reported that various dermatological disorders, such as age spots, melasma, post-inflammatory melanoderma, freckles, and sites of actinic damage, result from the accumulation of an excessive level of epidermal melanin [13]. Inhibitors of melanin synthesis have been increasingly applied in skin care products for the treatment or prevention of skin hyperpigmented disorders [14,15]. In addition, antioxidants, such as arbutin or kojic acid [16], have been used in the treatment of hyperpigmentation [17]. Recently, skin-whitening cosmetics have accounted for a large proportion of cosmetic products. Not only women with dark skin, but also those who are trying to cope with dark spots on their skin arising from UV light, pregnancy, or increased age widely make use of these skin-whitening products.

Melanogenesis is regulated by multiple factors. For example, tyrosinase related protein-1 (TRP1), microphthalmia-associated transcription factor (MITF), and tyrosinase-related protein-2 (TRP2) were reported to regulate the production of melanin [18,19,20]. The melanocortin 1 receptor (MC1R) also features prominently in α-MSH-induced melanin synthesis [21]. There are several signaling pathways involved in melanin production. Among the different signaling pathways regulating melanin production, cyclic AMP (cAMP)-mediated protein kinase A (PKA) activation plays a critical role in the regulation of melanogenesis [22]. In the cAM-PKA signaling pathway, cAMP induces the activation of PKA [23], which is followed by phosphorylation of cAMP response element-binding protein (CREB). CREB is an intracellular transcription factor that stimulates the expression of the microphthalmia factor (MITF) gene, which is important in melanogenesis. MITF is a transcription factor that binds to the promoter regions of the melanogenic genes tyrosinase, TRP1, and TRP2, upregulating their expression [24,25]. Afterwards, the intracellular melanin content increases [26]. It has been reported that α-MSH increases cAMP levels and is usually applied to activate the phosphorylation of CREB and then enhance MITF protein levels [27].

MITF expression is regulated by several signaling pathways. c-Jun N-terminal kinase (JNK) and p38 mitogen-activated protein kinase (MAPK) are involved in the activation of MITF expression and the consequent increased tyrosinase expression. Extracellular responsive kinase (ERK) activation signals also increase CREB phosphorylation and subsequent MITF expression, which modulates melanin synthesis [28,29]. Hence, several skin-whitening agents were reported to inhibit MITF transcriptional activity by decreasing the protein expression levels of tyrosinase, TRP1, and TRP2 through downregulation of p38MAPK-mediated MITF phosphorylation. In addition, the ERK pathway has been described to be involved in MITF regulation during melanogenesis [30,31]. Activation of ERK phosphorylates MITF, leading to degradation of MITF, resulting in a reduction of melanin content [32,33].

Oxidative stress can be the result of either ROS overproduction or decreased antioxidant defense [1]. The search for and applications of natural antioxidants remain in the focuses of numerous research teams all over the world. Recently, there has been growing interest in the research and development of phytochemicals, such as natural antioxidants and phenolic compounds from various natural sources, which could be used as nutraceuticals, anti-aging cosmetics, and medicinal products [34]. *Phoenix dactylifera* L. is one of the major fruit crops produced in the Middle East and North Africa [35,36]. Previous studies have reported that *P. dactylifera* L. fruits exhibit antioxidant, free radical-scavenging, antimicrobial, and anti-inflammatory activities [37,38]. The functional effects of *P. dactylifera* L. are not only restricted to the fruits themselves but also to their seeds, which are a kind of by-product of the fruits [39,40]. Owing to its antioxidant properties, *P. dactylifera* L. seed extracts could serve as a source of antioxidant substances that act against oxidative stress [41,42,43]. However, few studies have been conducted on the applications of *P. dactylifera* L. seeds for the development of cosmeceutical products. These seeds are usually discarded as waste, but their significance as a functional cosmetic ingredient is highlighted herein. To date, there have been no reports regarding the mechanism of action of *P. dactylifera* L. seed extract in the field of skin care cosmetics or the effects of the seed extract on melanin production. The focus of the present study was to determine the potential of *P. dactylifera* L. seed extract as a melanogenesis inhibitor for the cosmetic industry.

The mechanisms underlying the inhibitory effects of *P. dactylifera* L. seed extract on melanogenesis have not been investigated yet. The aim of the current study was to investigate the antioxidant characteristics and potential mechanisms of action of *P. dactylifera* L. seed extract on melanogenesis by examining the MITF transcription regulators and phosphorylation of regulators of the MAPK and PKA signaling pathways.

## 2. Materials and Methods

### 2.1. Chemicals and Reagents

The chemical reagents used in the study were purchased from Sigma-Aldrich Chemical Co. (St. Louis, MS, USA). The antibodies were from Santa Cruz Biotech (Santa Cruz, CA, USA) and the ECL reagent was from Millipore (Billerica, MA, USA).

### 2.2. Preparation and Extraction of P. dactylifera Seed Extract

Dried *P. dactylifera* L. fruits were harvested in 2019 from traditional stores located in Medina City, Saudi Arabia. The *P. dactylifera* L. seeds were washed completely, exposed to sunlight, air-dried for one day, and then dried at 80 °C for 2 h in an oven. The dehydrated seeds were pulverized to a fine powder (#50 mesh) with a committed mill (Retsch Ultra Centrifugal Mill and Sieving Machine, Type ZM1, Haan, Germany). The powder was collected in a sealed glass bottle and stored at 25 °C until use. Pulverized desiccated *P. dactylifera* L. seed powder (2 g) was placed in an extraction beaker containing 10 mL of distilled water. Next, ultrasonic extraction was carried out with the ULTRAsonik™ 57H model (250 W, C and A Sales Industrial Supplies, Yucaipa, CA, USA). The working frequency was 45 kHz for 30 min. After extraction, the samples were centrifuged at 4500 rpm for 50 min at 25 °C with an Eppendorf Centrifuge 5810 R (Hamburg, Germany). The supernatants were collected and filtered with a nylon filter (pore size: 0.45 µm) and the filtrate was concentrated to 9.8 mg/mL.

### 2.3. Ultra-Performance Liquid Chromatography (UPLC) Analysis of P. dactylifera Seed Extract and Ferulic Acid

The *P. dactylifera* seed extract was analyzed using an ACQUITY UPLC H class with a photodiode array detector (Waters, Milford, MA, USA). The column was an ACQUITY UPLC BEH C18 column, 130 Å, 1.7 µm, 2.1 mm × 100 mm. Mobile phase A was 0.5% acetic acid, and mobile phase B was acetonitrile. For time points 0 and 5 min, the percentage ratio of A/B was 95/5; for 20 min, the percentage ratio of A/B was 5/95; for time points 21 and 25 min, the percentage ratio of A/B was 95/5. Ferulic acid (1 ppm) was used as the positive standard.

### 2.4. DPPH Scavenging Activity Assay

In this assay, vitamin C (0.5 mg/mL) and BHA (0.1 mg/mL) were used as antioxidant standards. The *P. dactylifera* L. seed extract at various concentrations (0.0049, 0.0245, and 0.049 mg/mL) was added to 2.9 mL of DPPH (60 μM) solution. Antioxidant components in the seed extract could donate hydrogen to DPPH and convert DPPH to the reduced form. The resulting decrease in absorbance at 517 nm was recorded using a UV-Vis spectrophotometer [44].

### 2.5. ABTS^+^ Scavenging Capacity Assay

The ABTS^+^ scavenging capacity of *P. dactylifera* L. seed extract was compared with those of vitamin C (0.9 mg/mL) and BHA (0.9 mg/mL). For this assay, ABTS^+^ was used. Cation was first produced by reacting the solution of ABTS (7 mM) with potassium persulfate (2.45 mM), and the mixture was allowed to stand in the dark for at least 6 h before use. The absorbance at 734 nm was measured 10 min after mixing of different concentrations of the seed extract (0.0098, 0.049, and 0.098 mg/mL) with 1 mL of ABTS^+^ solution [45].

### 2.6. Determination of Total Phenolic Content

The total phenolic content was determined using the Folin–Ciocalteu reagent [46], and gallic acid (2 μg/mL) was employed as a positive standard. Different concentrations of *P. dactylifera* L. seed extract (0.0196, 0.098, and 0.196 mg/mL) were prepared in 80% methanol; 100 μL of extract was transferred into a test tube and 0.5 mL of Folin–Ciocalteu reagent (previously diluted 10-fold with deionized water) were added and mixed. The mixture was allowed to stand at room temperature for 5 min; 1.5 mL of 20% sodium carbonate was added. After standing at room temperature for 2 h, the absorbance was read at 760 nm using a UV–Vis spectrophotometer.

### 2.7. Determination of Reducing Capacity

Vitamin C (0.008 mg/mL) or BHA (4.8 mg/mL) were utilized as positive standards in this assay. The reducing capacity of the seed extract was determined according to the method described by Oyaizu [47]. Various concentrations of *P. dactylifera* L. seed extract (0.0073, 0.036, 0.073 mg/mL) were mixed with phosphate buffer (2.5 mL, 0.2 M, pH 6.6) and potassium ferricyanide [K_3_Fe (CN)_6_] (2.5 mL, 1% *w*/*v*). The mixture was incubated at 50 °C for 20 min. Trichloroacetic acid (2.5 mL, 10% *w*/*v*) was added to the mixture, which was then centrifuged at 1000× *g* for 10 min. The upper layer of solution (2.5 mL) was mixed with distilled water (2.5 mL) and FeCl3 (0.5 mL, 0.1% *w*/*v*), and the absorbance at 700 nm was measured.

### 2.8. Measurement of Iron (II) Ion Chelating Capacity

For the measurement of the Fe^2+^-ion chelating capacity of *P. dactylifera* L. seed extract, EDTA (0.02, 0.04, and 0.08 mg/mL) was used as a positive standard. Different concentrations of the seed extract (0.0196, 0.098, and 0.196 mg/mL) were added to a FeCl_2_ solution (0.05 mL, 1 mM). The reaction mixture was reacted with ferrozine (0.1 mL, 1 mM), then the mixture was quantified to 1 mL with methanol and incubated at 25 °C for 10 min. The absorbance of the reaction mixture was measured at 562 nm [48].

### 2.9. Cell Viability Assay

The cell viability assay was performed using the MTT method [49]. The cells were exposed to various concentrations of *P. dactylifera* L. seed extract (0.049, 0.245, and 0.49 mg/mL) for 24 h at 37 °C and 5% CO_2_ in a humidified incubator, and the MTT solution was added to the wells. The insoluble derivative of MTT was solubilized with ethanol-DMSO (1:1 mixture solution). The absorbance of the wells at 570 nm was measured using a microplate reader. B16F10 cells (BCRC60031) were obtained from the Bioresource Collection and Research Center (BCRC), Hsinchu city, Taiwan. The cells were maintained in DMEM (Hyclone, Logan, UT, USA) supplemented with 10% fetal bovine serum and 1% antibiotics. In addition, the effect of *P. dactylifera* L. seed on B16F10 cell viability was also evaluated by the trypan blue exclusion assay method. After treatment with the seed extract, cells were trypsinized and centrifuged to collect pellet. The cell pellet was re-suspended in 300 μL of DMEM cell culture media. A small aliquot of cell suspension (20 μL) was mixed gently with 10 μL trypan blue (4% in PBS). The cell number was counted (nonviable cells were blue and viable cells unstained) by using a hemocytometer under the microscope.

### 2.10. Measurement of Mushroom Tyrosinase Activity

The mushroom tyrosinase solution (10 μL, 200 units) was added to a 96-well microplate. The reaction mixture contained 5 mM of L-DOPA dissolved in phosphate buffered saline (PBS) (50 mM, pH 6.8), and *P. dactylifera* L. seed extract (0.049, 0.245, and 0.49 mg/mL), kojic acid (200 mM; 0.03 mg/mL), or arbutin (2 mM; 0.54 mg/mL). The assay mixture was incubated at 37 °C for 30 min and the absorbance of dopachrome produced was measured at 490 nm. Measurement of mushroom tyrosinase activity was conducted as previously described [50].

### 2.11. Determination of Melanin Content

The B16F10 cells were treated with α-MSH (100 nM) for 24 h, and then treated with either *P. dactylifera* L. seed extract (0.0245–0.147 mg/mL) or arbutin (0.54 mg/mL) for an additional 24 h. After treatment, the cell pellets were solubilized in NaOH solution (1 N) at 60 °C for 60 min. The melanin content was detected at 405 nm, as previously described by Tsuboi [51].

### 2.12. Measurement of Intracellular Tyrosinase Activity

Intracellular tyrosinase activity was determined as described previously [52]. The cells were treated with α-MSH (100 nM) for 24 h and then with *P. dactylifera* L. seed extract (0.0245–0.147 mg/mL) or arbutin (0.54 mg/mL) for 24 h. After treatment, the cell extracts (100 μL) were mixed with freshly prepared L-DOPA solution (0.1% in PBS) and incubated at 37 °C, and the absorbance at 490 nm was measured.

### 2.13. Western Blotting Experiment

The cells were treated with *P. dactylifera* L. seed extract (0.0245, 0.0368, 0.049, and 0.147 mg/mL) or arbutin (0.54 mg/mL) and then lysed in PBS containing nonidet P—40 (1%), sodium deoxycholate (0.5%), sodium dodecyl sulfate (SDS, 0.1%), aprotinin (5 μg/mL), phenylmethylsulfonyl fluoride (100 μg/mL), pepstatin A (1 μg/mL), and EDTA (1 mM) at 4 °C for 20 min. Total lysates were quantified using a microBCA kit (Thermo Fisher Scientific, Waltham, MA, USA). Proteins (30 μg) were resolved by SDS-polyacrylamide gel electrophoresis and electrophoretically transferred to a polyvinylidene fluoride membrane. The membrane was blocked in 5% fat-free milk in PBST (PBS with 0.05 % Tween-20) and incubated at 4 °C overnight with the following primary antibodies diluted in PBST: MITF Ab (1:1000), TRP1 Ab (1:6000), TRP2 Ab (1:1000), MC1R Ab (1:500), GAPDH Ab (1:1500), tyrosinase Ab (1:2000), p-p38 Ab (1:500), p38 Ab (1:500), p-JNK Ab (1:500), JNK Ab (1:500), p-ERK Ab (1:500), ERK Ab (1:500), p-CERB Ab (1:500), and CERB Ab (1:200), all from Santa Cruz Biotech (Dallas, TX, USA)). The bound antibodies were detected by horseradish peroxidase-conjugated secondary antibody (Amersham Corp.) followed by ECL detection system (Amersham) according to the manufacturer’s instruction.

### 2.14. PKA Inhibitor Assay

The cells were treated with α-MSH (100 nM) for 24 h followed by a 1 h addition of 10 μM of PKA regulator H89. After treatment, *P. dactylifera* L. seed extract (0.147 mg/mL) and H89 was added to the cells and incubated for an additional 23 h. The melanin content was determined as described above.

### 2.15. Determination of Intracellular ROS Level

The B16F10 melanoma cells were seeded in 24-well plates (5 × 10^4^ cells/mL) and treated with various concentrations of *P. dactylifera* L. seed extract (0.049, 0.245, and 0.49 mg/mL) and Trolox^®^ (0.05 mg/mL) for 24 h. After treatment with 24 mM H_2_O_2_ at 37 °C for 30 min, 2′ 7′-dichloro-fluorescein diacetate (DCFH-DA; Sigma-Aldrich) (10 μM in PBS) was added to each well for 30 min. The medium was discarded, and the cells were washed twice with PBS. The ROS fluorescence was monitored by fluorescence microscopy (Olympus IX71). ROS fluorescence was monitored using a fluorescent reader, Fluoroskan Ascent (Thermo Scientific, Vantaa, Finland) and analyzed with Ascent software (Thermo Scientific, Vantaa, Finland) [53].

### 2.16. RT-PCR

Quantitative PCR was carried out on an Applied Biosystems 7300 Real-Time PCR system as follows: 95 °C for 10 min, 40 cycles of 95 °C (15 s), and 60 °C (1 min) using 2× Power SYBR Green PCR Master Mix (Applied Biosystems, Foster City, CA, USA), which contains forward and reverse primers (200 nM) (tyrosinase gene forward primer: TTGCCACTTCATGTCATCATAGAATATT; tyrosinase gene reverse primer: TTTATCAAAGGTGTGACTGCTATACAAAT; TRP1 gene forward primer: ATGCGGTCTTTGACGAATGG; TRP1 gene reverse primer: CGTTTTCCAACGGGAAGGT; TRP2 gene forward primer: CTCAGAGCTCGGGCTCAGTT; TRP2 gene reverse primer: TGTTCAGCACGCCATCCA; MITF gene forward primer: CGCCTGATCTGGTGAATCG; MITF gene reverse primer: CCTGGCTGCAGTTCTCAAGAA; GAPDH gene forward primer: CGTCCCGTAGACAAAATGGT; GAPDH gene reverse primer: TTGATGGCAACAATCTCCAC).

### 2.17. Statistical Analysis

Statistical analysis was carried out using Tukey’s post hoc test, which was used for comparisons of measured data, using the Statistical Package for the Social Sciences (SPSS) version 22.0 statistical software (SPSS Inc., Chicago, IL, USA). * *p* < 0.05 was considered significant, ** *p* < 0.01, *** *p* < 0.001.

## 3. Results

### 3.1. UPLC Analysis of P. dactylifera L. Seed Extract

The UPLC analysis results are shown in Figure 1A,B. Figure 1A depicts the overlapped peak of *P. dactylifera* L. seed extract and ferulic acid at 330 nm. Figure 1B shows similar scan spectra between 220 and 500 nm for *P. dactylifera* L. seed extract and ferulic acid.

### 3.2. Antioxidant Characteristics of P. dactylifera L. Seed Extract

The antioxidant activity of *P. dactylifera* L. seed extract in vitro revealed the presence of antioxidant potential. In DPPH assay, vitamin C (0.05 mM; 0.53 mg/mL) and tert-butyl hydroxyanisole (BHA) (0.1 mg/mL) were used as positive antioxidant standards. The DPPH scavenging capacity of the extract was 49.97 ± 2.9%, 81.36 ± 0.56%, and 78.53 ± 3.83% of the control for the extract concentrations of 0.0049, 0.0245, and 0.049 (mg/mL), respectively. In comparison, the scavenging capacities of vitamin C and BHA were 90.12 ± 0.31% and 90.44 ± 0.49%, respectively (Figure 2A).

This ABTS radical cation is blue in color and is reactive towards most antioxidants. During this reaction, the blue ABTS radical cation is converted back to its colorless neutral form. The reaction was monitored spectrophotometrically. The ABTS^+^ scavenging capacity of the extract was 5.69 ± 1.36%, 18.81 ± 0.68%, and 66.82 ± 8.51% of the control at concentrations of 0.0098, 0.049, and 0.098 mg/mL, respectively. In contrast, the ABTS^+^ scavenging capacities of vitamin C (0.9 mg/mL) and BHA (0.9 mg/mL) were 95.52 ± 0.12% and 40.3 ± 0.83%, respectively. The results in Figure 2B indicate that the seed extract scavenges a significant amount of the ABTS^+^ radical.

To determine the total phenolic content of the *P. dactylifera* L. seed extract, gallic acid (2 μg/mL) was used as a positive standard. The results in Figure 2C indicate that the total phenolic contents in 0.0196, 0.098, and 0.196 (mg/mL) of the extract were 33.43 ± 2.33%, 117.22 ± 7.81%, and 195.4 ± 10.91%, respectively. The gallic acid equivalents (GAE) of the aforementioned concentrations of seed extracts were 0.068 ± 0.01, 0.365 ± 0.01, and 0.642 ± 0.03, respectively (Figure 2C).

To determine the reducing capacity of the seed extract, various concentrations of the extract (0.0073, 0.036, and 0.073, mg/mL), vitamin C (0.008 mg/mL), and BHA (4.8 mg/mL) were tested. The results depicted in Figure 2D reveal that higher concentrations of the extract present apparent reducing capacity. The reducing capacities of 0.0073, 0.036, and 0.073 mg/mL of seed extract were 18.82 ± 2.3%, 52.72 ± 2.41%, and 88.33 ± 4.89%, respectively. In addition, the reducing capacities of vitamin C (0.008 mg/mL) and BHA (4.8 mg/mL) were 102.87 ± 3.02% and 100 ± 0.01%, respectively.

The *P. dactylifera* L. seed extract featured lower metal ion-chelating characteristics than the positive standard, EDTA. The metal ion-chelating capacities of the seed extract at 0.098 and 0.196 mg/mL were 6.82 ± 2.69% and 19.78 ± 3.09%, respectively. EDTA exhibited a higher metal ion-chelating capacity than the seed extract. For 0.02, 0.04, and 0.08 (mg/mL) of EDTA, the metal ion chelating capacities were 36.02 ± 8.21%, 60.05 ± 4.56%, and 98.3 ± 0.07%, respectively (Figure 2E).

After treatment with H_2_O_2_ (20 mM), the cellular ROS induced by H_2_O_2_ was 59.19 ± 7.47% and 76.96 ± 7.14% for 0.245 and 0.49 mg/mL of the *P. dactylifera* L. seed extract, respectively (Figure 2F).

### 3.3. Cell Viability Assay

The effect of *P. dactylifera* L. seed extract on B16F10 cell viability was determined by the MTT assay method. The cells were treated with various concentrations of the seed extract (0.049, 0.245, and 0.49 mg/mL) for 24 h and then the MTT assay was performed. Results are expressed as percentages of viability relative to the control. After treatment, the cell viabilities were 97.99 ± 1.1%, 98.41 ± 2.9%, and 98.56 ± 1.24% for 0.049, 0.245, and 0.49 mg/mL of the seed extract, respectively, compared with that of the control cells (Figure 3A). For trypan blue exclusion assay, the cell viabilities were 100.07 ± 1.42%, 100.01 ± 0.21%, and 100.01 ± 0.22% for 0.049, 0.245, and 0.49 mg/mL of the seed extract, respectively. The results indicated that the *P. dactylifera* L. seed extract showed no cytotoxic effect on B16F10 cell viability.

### 3.4. Inhibitory Effects of P. dactylifera L. Seed Extract on Melanogenesis

The results shown in Figure 4A demonstrate that 0.049 mg/mL of *P. dactylifera* L. seed extract did not exert an inhibitory effect on mushroom tyrosinase activity. On the other hand, the enzyme inhibition percentages were 8.34 ± 2.67% and 33 ± 2.36% of the control for the 0.245 and 0.49 mg/mL *P. dactylifera* L. seed extract treatments, respectively. In addition, the tyrosinase inhibition percentages of kojic acid (200 μM; 0.03 mg/mL) and arbutin (2 mM; 0.54 mg/mL) were 56.26 ± 5.06% and 64.56 ± 2.88% of the control, respectively (Figure 4A). Thus, the higher concentrations of *P. dactylifera* L. seed extract (0.245 and 0.49 mg/mL) could represent inhibitory levels of mushroom tyrosinase.

In Figure 4B, the results indicated that only 0.147 mg/mL of *P. dactylifera* L. seed extract could significantly decrease melanin content in B16F10 melanoma cells. After treatment, the melanin content in B16F10 cells was 80.66 ± 5.43% of that of the control. For the positive standard, arbutin (0.54 mg/mL), the remaining intracellular melanin content was 61.19 ± 8.17% of that of the control. The results indicated that 0.147 mg/mL of *P. dactylifera* L. seed extract showed smaller effects than arbutin. The remaining intracellular tyrosinase activity values were 85.1 ± 8.1% and 73.7 ± 11.9% for the 0.098 and 0.147 mg/mL of *P. dactylifera* L. seed extract, respectively. The remaining intracellular tyrosinase activity was 64.3 ± 14.9% of the control after the cells were treated with arbutin (Figure 4C). The results indicated that higher concentrations of *P. dactylifera* L. seed extract still exhibited a smaller inhibitory effect on α-MSH-induced tyrosinase activity in B16F10 cells than arbutin.

### 3.5. P. dactylifera L. seed Extract Inhibits the Expression Levels of Proteins Involved in Melaninogenesis

Western blots were used to examine the expression levels of melanogenesis-related proteins in B16F10 cells (Figure 5A). The results indicate that treatment with various concentrations of *P. dactylifera* L. seed extract led to reduced levels of MC1R, MITF, tyrosinase, TRP1, and TRP2. The inhibitory effects of the extract on protein expression were apparent at a concentration of 0.147 mg/mL. The fold change in protein expression level for MC1R was 0.74 ± 0.02; for MITF it was 0.51 ± 0.03; for tyrosinase it was 0.54 ± 0.26; for TRP-1 it was 0.57 ± 0.15; and for TRP-2 it was 0.49 ± 0.1 (Figure 5B).

The expression levels of melanogenesis-related signaling proteins were examined using Western blots (Figure 5C). The results indicate that 0.147 mg/mL of *P. dactylifera* L. seed extract treatment apparently led to reduced levels of p-p38, p-JNK, p-ERK, and p-CREB. The fold change of protein expression level for p-p38 was 0.88 ± 0.15; for p-JNK it was 0.68 ± 0.39; for p-ERK it was 0.65 ± 0.23; and for p-CREB it was 0.44 ± 0.07 (Figure 5D).

### 3.6. P. dactylifera L. Seed Extract Does Not Affect Tyrosinase, TRP1, TRP2, or MITF Gene Expression

The gene expression levels of melanogenesis-related proteins, such as tyrosinase, TRP1, TRP2, and MITF were examined by quantitative real-time polymerase chain reaction (RT-PCR). The relative normalized expression levels of tyrosinase/GAPDH for 0.0367, 0.049, and 0.147 mg/mL of *P. dactylifera* L. seed extract treatments were 3.51 ± 0.52, 2.38 ± 0.24, and 1.64 ± 0.55, respectively. For TRP-1/GAPDH, the relative normalized expression levels for 0.0367, 0.049, and 0.147 mg/mL of *P. dactylifera* L. seed extract treatments were 1.64 ± 0.2, 1.15 ± 0.02, and 0.89 ± 0.07, respectively. The relative normalized expression levels of TRP-2/GAPDH for 0.0367, 0.049, and 0.147 mg/mL of *P. dactylifera* L. seed extract were 1.08 ± 0.13, 0.85 ± 0.01, and 0.7 ± 0.05, respectively. For MITF/GAPDH, the relative normalized expression levels for 0.0367, 0.049, and 0.147 mg/mL of *P. dactylifera* L. seed extract treatments were 1.48 ± 0.19, 1.03 ± 0.02, and 0.74 ± 0.05, respectively (Figure 6).

## 4. Discussion

It has been reported that melanogenesis increases cellular oxidative stress. In addition, several antioxidants, ROS scavengers and inhibitor scavengers, and inhibitors may inhibit UV-mediated melanin synthesis [54]. Therefore, antioxidants, ROS scavengers, and inhibitors of melanogenesis have been increasingly applied to skin-whitening cosmetics for the prevention of undesirable skin hyperpigmentation [14]. It was also found that stimulation of metallothionein, an endogenous antioxidant could suppress melanogenesis in melanocytes [55]. Human skin is frequently exposed to environmental oxidizing pollutants or UV light and is damaged by extrinsic environmental factors. In particular, UV irradiation has been reported to induce generation of ROS in skin and promote cell membrane lipid peroxidation [56]. To counteract oxidative stress, skin is equipped with particular antioxidant systems [57].

To elucidate the antioxidant activity of *P. dactylifera* L. seed extract, DPPH and ABTS^+^ radical scavenging activity, total phenolic content, reducing capacity, and metal ion-chelating capacity of the extract were determined as previously described [47,48]. The *P. dactylifera* L. seed extract exerted considerable antioxidant activities in all of the aforementioned analytical studies. The results showed the antioxidant potential of *P. dactylifera* L. seed extract over different ranges with distinct efficiencies. The DPPH-scavenging activity of the seed extract shown in Figure 2A was different from that of ABTS^+^ shown in Figure 2B, which may have resulted from different mechanisms of the antioxidant–radical interactions. In addition, the stoichiometry of reactions between the antioxidant components in the seed extract may vary, which results in a difference in free radical-scavenging capacity [58]. The potential antioxidants in the *P. dactylifera* L. seed extract could convert a Fe^3+^/ferricyanide complex to the ferrous form, and the reducing capacity served as an indicator of the antioxidant activity of the seed extract shown in Figure 2D. It was found that kojic acid [59] acts as a tyrosinase inhibitor because kojic acid acts as a metal chelator to chelate the copper ions and inhibit those ions entering the active site of tyrosinase. Hence, antioxidants in the seed extract may form insoluble metal complexes with ferrous ions and then inhibit the interaction between metal ions and lipids. The higher metal ion-chelating capacity of the seed extract indicated its potential antioxidant activity and possible inhibitory effects on tyrosinase activity (Figure 2E).

The principle of the assay of intracellular ROS is that DCFH-DA diffuses through the cell membrane and is enzymatically hydrolyzed to DCFH by esterase; then DCFH reacts with ROS (such as H_2_O_2_) to yield DCF. Rapid increases in DCF indicated the oxidation of DCFH by intracellular radicals [60]. The idea behind searching novel antioxidants for skin-whitening activities lies in the hypothesis that the oxidative stress resulting from UV irradiation may contribute to the stimulation of melanin synthesis. UV irradiation has been reported to produce ROS in cutaneous tissues that may induce melanin production by activating tyrosinase [61]. Moreover, it was reported that several skin-whitening agents can inhibit melanogenesis by interacting with copper ions at the active site of tyrosinase or with *o*-quinones to block the polymerization of intermediates in the melanin synthetic pathway [62]. Additionally, vitamin C or vitamin E can reduce the oxidation of pre-existing melanin particles. Hence, these vitamins have been widely applied in skin-whitening products [63]. Interestingly, our results demonstrated the antioxidant characteristics of *P. dactylifera* L. seed extract, indicating that the seed extract of *P. dactylifera* L. could act as a skin-whitening ingredient in cosmetics.

The MTT assay is a well-known colorimetric assay that measures the activity of cellular NADH/NADPH-dependent oxidoreductase enzymes that reduce MTT to purple-colored formazan dyes. This assay could be applied to examine the potential cytotoxicity of medicinal agents and toxic materials, as these agents enhance or inhibit cell viability. The results shown in Figure 3A were in accordance with those of the Trypan blue exclusion assay in Figure 3B, which indicate that the *P. dactylifera* L. seed extract had no cytotoxic effect on B16F10 melanoma cell viability.

Mushroom tyrosinase is commonly used as a target enzyme for screening potential inhibitors of melanogenesis. It was first found that the dosage range (0.049–0.49 mg/mL) of the *P. dactylifera* L. seed extract could inhibit the activity of mushroom tyrosinase. The results shown in Figure 4A indicate that the seed extract exhibited lower inhibitory impacts on mushroom tyrosinase activity than kojic acid. Tyrosinase plays a major role in the first two steps of the melanogenesis pathway. To elucidate the true inhibitory effect of the *P. dactylifera* L. seed extract on melanin production, the B16F10 melanin content and intracellular tyrosinase activity were determined. The results presented in Figure 4B indicate that a higher concentration of *P. dactylifera* L. seed extract exerted an apparent inhibitory effect on melanin formation. The data showed that *P. dactylifera* L. seed extract truly blocks melanogenesis in melanoma cells. With this, the results featured in Figure 4C were in accordance with the results shown in Figure 4B. In the aforementioned cellular experiments, α-MSH was used as an inducer to stimulate melanin synthesis. α-MSH was reported to bind melanocortin 1 receptor (MC1R) and activate intracellular adenylate cyclase, which in turn catalyzes ATP to cAMP and activates cAMP-dependent PKA. The results in Figure 5A revealed that the *P. dactylifera* L. seed extract inhibited the expression of MC1R, thereby blocking melanin production induced by α-MSH mediated intracellular cAMP upregulation. Furthermore, the cAMP-mediated PKA signaling pathway was inactivated and melanogenesis was inhibited. To confirm the inference, we have carried the PKA inhibitor experiments, and the data shown in Figure 4D indicated that the inhibitory effect of *P. dactylifera* L. seed extract (0.147 mg/mL) on melanogenesis in B16F10 cells was suppressed by H89-treatment. H89 (N-[2-p-bromocinnamylamino-ethyl]-5-isoquinolinesulphonamide) is a selective and potent inhibitor of PKA. The results indicate that cAMP-mediated PKA signaling was affected by *P. dactylifera* L. seed extract.

Tyrosinase, TRP1, and TRP2 are the three major enzymes that regulate melanin synthesis in mammalian cells [64]. Furthermore, MITF is the major transcriptional regulator of tyrosinase, TRP1, and TRP2, and is the most important regulator of melanocyte pigmentation [65]. The results in Figure 5A indicate that the *P. dactylifera* L. seed extract decreased the protein expression levels of these melanogenesis-related proteins, inhibited tyrosinase activity, and decreased melanin content in B16F10 cells.

In this study, *P. dactylifera* L. seed extract suppressed CREB phosphorylation and PKA activation. These data suggest that the cAMP-PKA pathway may be responsible for *P. dactylifera* L. seed extract-induced anti-melanogenesis (Figure 5C).

MAPKs have been reported to modulate melanogenesis [66]. The MAPK family comprises three types of protein kinases, including extracellular responsive kinase (ERK), c-Jun N-terminal kinase (JNK), and p38 MAPK. The p38 MAPK can activate the cAMP response element-binding protein (CREB, and CREB activates MITF expression, which positively contributes to melanin synthesis [67]. The results in Figure 5C provide evidence that the *P. dactylifera* L. seed extract could inactivate CREB, JNK, and p38, in turn inhibiting MITF expression (Figure 5A). These results suggest that the *P. dactylifera* L. seed extract-induced antimelanogenic effect may be mediated by downregulation of the PKA pathways. Moreover, UV light irradiation has been reported to playfeature prominently in the initiation of several skin disorders, including scaling, wrinkling, and hyperpigmentation [68]. The results suggested that *P. dactylifera* L. seed extract decreased melanin production, which may be attributed to the depletion of intracellular ROS.

To assess the effect of *P. dactylifera* L. seed extract on cellular melanin production, we determined the mRNA levels of in B16F10 cells treated with α-MSH and *P. dactylifera* L. seed extract. The α-MSH-treated cells showed increased levels of MITF, tyrosinase, TRP-1, and TRP-2, as expected. Interestingly, *P. dactylifera* L. seed extract-treated cells showed a reduction in the mRNA expression of these melanogenesis-related genes. However, no statistically significant difference existed among columns, which indicated that *P. dactylifera* L. seed extract does not affect the gene expression levels of MITF, tyrosinase, TRP1, and TRP2.

It has been reported that the major phenolic acid found in *P. dactylifera* L. seed is ferulic acid [69]. Ferulic acid has been shown to effectively inhibit melanin synthesis in B16 melanoma cells by inhibiting casein kinase 2-induced phosphorylation of tyrosinase in a dose-dependent manner [70]. These reports support our results shown in Figure 1—ferulic acid in the water extract of *P. dactylifera* L. seed contributed to the inhibition of melanogenesis in B16F10 cells. Certainly, other functional components in the seed extract should be investigated in the near future.

Our results indicated that *P. dactylifera* L. seed extract inhibited melanogenesis in B16F10 cells by downregulating PKA signaling pathways. Hence, the *P. dactylifera* L. seed extract could be used as an effective skin-whitening agent.

## 5. Conclusions

This is the first report on the mechanism of action of *P. dactylifera* L. seed water extract in melanin synthesis. The present study demonstrated the involvement of cAMP/PKA/CREB pathway in the depigmenting effect of *P. dactylifera* L. seed extract. Our results indicated that the *P. dactylifera* L. seed extract inhibited melanogenesis in B16F10 cells by downregulating the PKA signaling pathways. Hence, the *P. dactylifera* L. seed extract could be used as a novel dermatological antimelanogenesis agent and an effective skin-whitening agent.

## Figures and Tables

**Figure 1 antioxidants-09-01270-f001:**
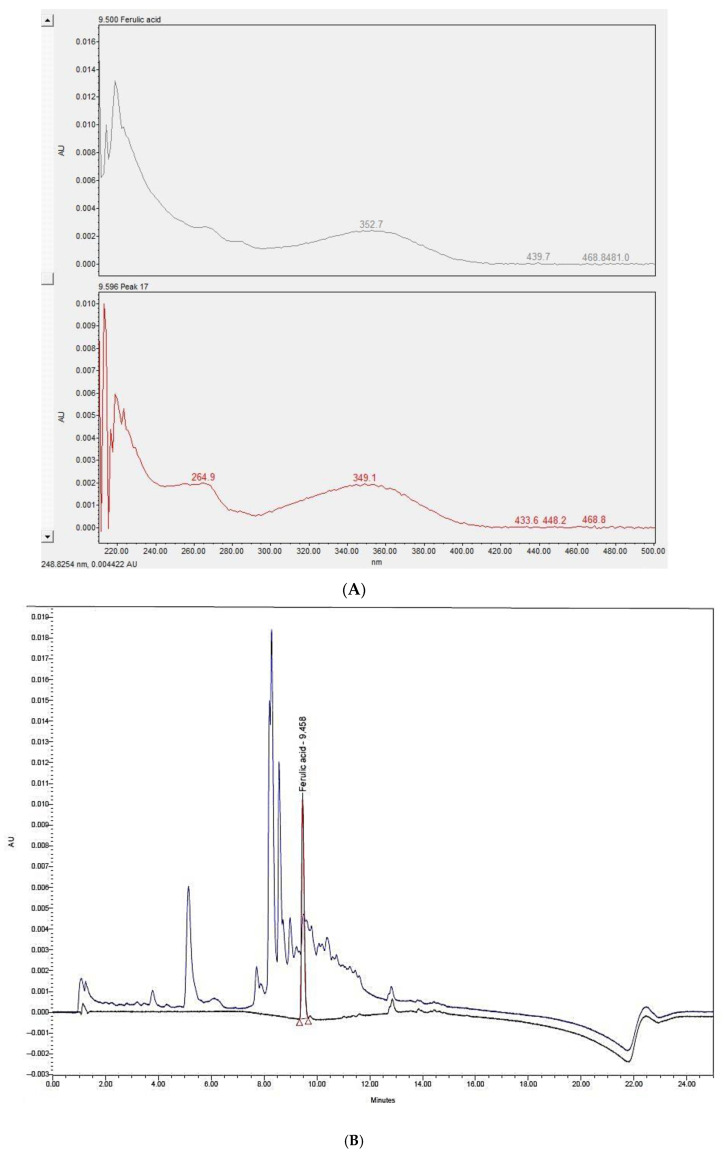
UPLC analysis of *Phoenix dactylifera* L. seed extract. (**A**) The absorbance at 330 nm of *P. dactylifera* L. seed extract (blue peak) and ferulic acid (black peak). (**B**) Scan spectra between 220 and 500 nm of *P. dactylifera* L. seed extract and ferulic acid.

**Figure 2 antioxidants-09-01270-f002:**
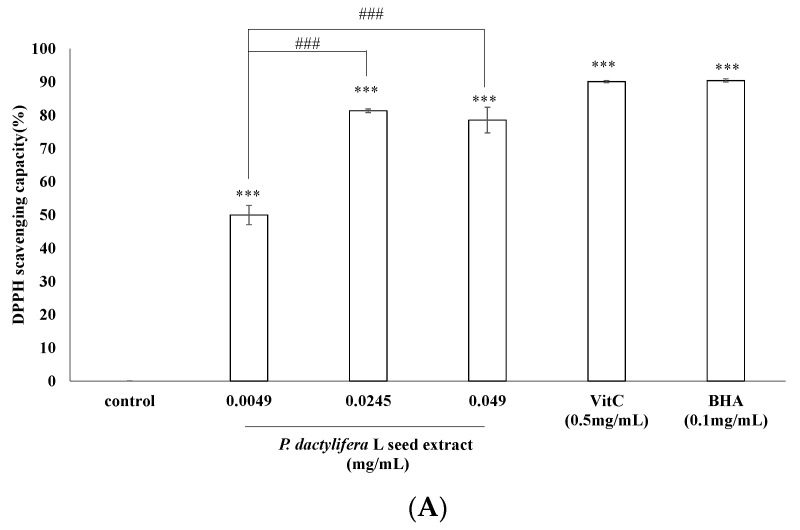
Antioxidant characteristics of *P. dactylifera* L. seed extract. (**A**) DPPH assay. The extract at various concentrations (0.0049, 0.0245, and 0.049 mg/mL), vitamin C (0.53 mg/mL), or BHA (0.1 mg/mL) interacted with DPPH. The control was DPPH only. (**B**) ABTS^+^ assay. The ABTS^+^ scavenging capacity of the extract (0.0098, 0.049, and 0.098 mg/mL) was compared with those of vitamin C (0.9 mg/mL) and BHA (0.9 mg/mL). (**C**) Total phenolic content assay. Different concentrations of extracts were tested with the Folin–Ciocalteu reagent. The absorbance of samples was measured at 760 nm. (**D**) Reducing capacity assay. The absorbance at 700 nm of different concentrations of extract (0.0073, 0.036, and 0.073 mg/mL), vitamin C (0.008 mg/mL), and BHA (4.8 mg/mL) were measured. (**E**) Ferrous ion-chelating capacity assay. Different concentrations of extract (0.0196, 0.098, and 0.196 mg/mL) or the positive standard EDTA (0.02, 0.04, and 0.08 mg/mL) were added to a reaction solution. The absorbance of the reaction mixture was measured at 562 nm. (**F**) ROS assay. The B16F10 melanoma cells were pretreated with various concentrations of *P. dactylifera* L. seed extract (0.049, 0.245, and 0.49 mg/mL), Trolox^®^ (0.05 mg/mL), or nothing for 24 h. The cells were then incubated with 24 mM H_2_O_2_ and DCFH-DA, and the fluorescence intensities of DCF were measured; the ROS levels were also calculated. Results are represented as percentages of control, and the data are mean ± SD for three separate experiments. Values are significantly different by comparison with control. ** *p* < 0.01, *** *p* < 0.001; ## *p* < 0.05, ### *p* < 0.001 seed extract group comparison; ψ, *p* < 0.05 seed extract (0.098 mg/mL) vs. (0.196 mg/mL).

**Figure 3 antioxidants-09-01270-f003:**
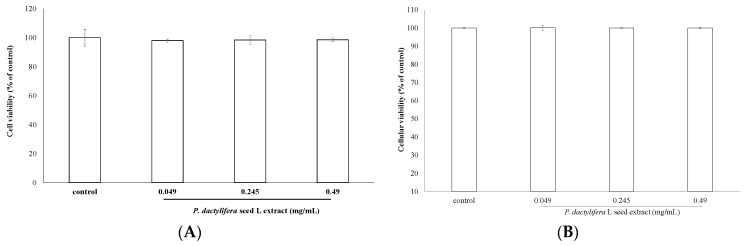
The effect of *P. dactylifera* L. seed extract on the proliferation of B16F10 cells. Cell viability was measured by the MTT assay method (**A**) and Trypan blue exclusion assay (**B**) after 24 h incubation. Data are expressed as percentages of the number of viable cells observed with the control and each column was presented as mean values ± SD from three independent experiments performed in triplicate.

**Figure 4 antioxidants-09-01270-f004:**
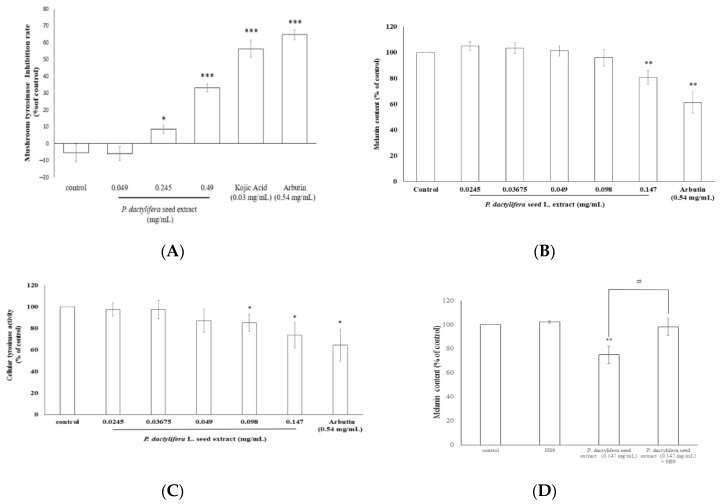
The inhibitory effects of *P. dactylifera* L. seed extract on melanogenesis. (**A**) The effects of *P. dactylifera* L. seed extract on mushroom tyrosinase activity. (**B**) The effects of *P. dactylifera* L. seed extract on melanin content in B16F10 cells. (**C**) The effects of *P. dactylifera* L. seed extract on B16F10 tyrosinase activity. (**D**) The PKA inhibitor (H89) assay. The values are significant as compared with the control. * *p* < 0.05, ** *p* < 0.01, *** *p* < 0.001, # *p* < 0.05 seed extract (0.147 mg/mL) vs. seed extract (0.147 mg/mL) + H89.

**Figure 5 antioxidants-09-01270-f005:**
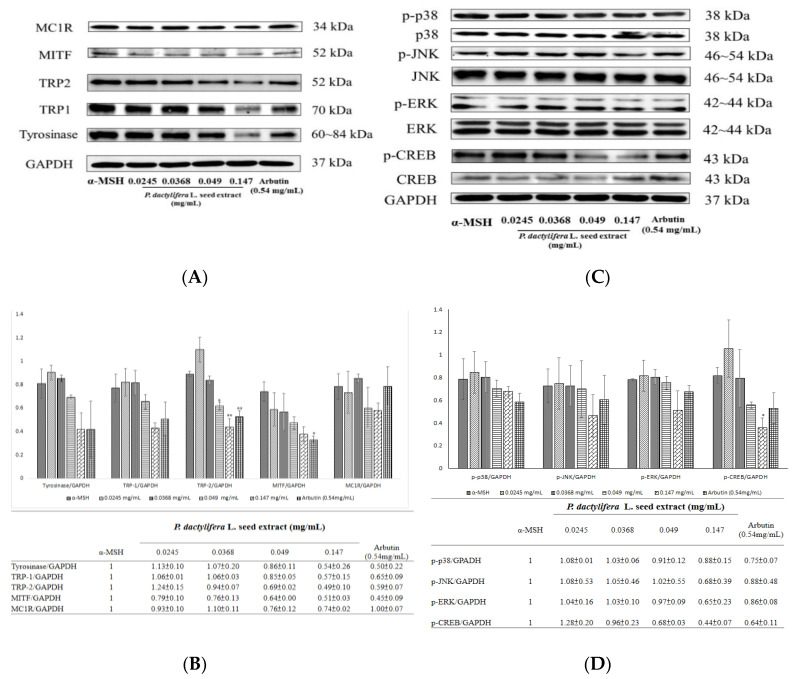
The effects of *P. dactylifera* L. seed extract on melanogenesis-related protein expression and signaling pathways. (**A**,**C**) Western blotting of cellular proteins in B16F10 cells. (**B**,**D**) The relative amounts of MITF, tyrosinase, MC1R, TRP1, and TRP2 or phosphorylated proteins (p-p38, p-JNK, p-ERK, and p-CREB) compared to total GAPDH were calculated and analyzed using Multi Gauge 3.0 software, and the values are represented as the means of triplicate experiments with standard deviations. * *p* < 0.05, ** *p* < 0.01.

**Figure 6 antioxidants-09-01270-f006:**
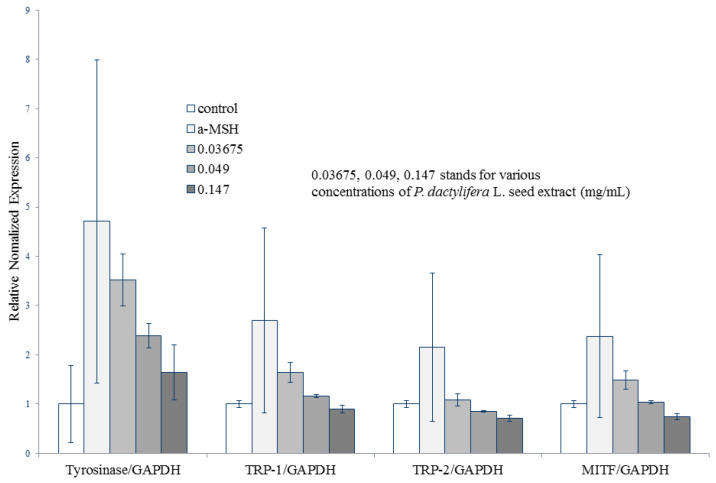
The effects of *P. dactylifera* L. seed extract on melanogenesis-related gene expression levels. Agarose gel electrophoresis of RT-PCR products of MITF, tyrosinase, TRP1, and TRP2. Total RNA from B16F10 cells treated with *P. dactylifera* L. seed extract (40 mg/mL) collected at the indicated time points. The mRNA levels were examined by real-time RT-PCR using glyceraldehyde 3-phosphate dehydrogenase (GAPDH) as an internal control.

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
