# Peer review of "Phoenix dactylifera* L. Seed Extract Exhibits Antioxidant Effects and Attenuates Melanogenesis in B16F10 Murine Melanoma Cells by Downregulating PKA Signaling"

_antioxidants, 2020, doi:10.3390/antiox9121270_

Round 1
Reviewer 1 Report
In this manuscript, the authors are aimed at assessing the prospective antioxidant and anti-melanogenic properties of P. dactylifera seed extracts in B16F10 melanoma cells, a widely employed cell model to study oxidative stress and melanogenesis. There are several shortcomings that should be addressed:
- English language is fine, however the manuscript should be carefully checked by a native speaker colleague as some grammar and typographical errors are present.
- The authors do not mention which post-hoc test is used to calculate statistical significance between columns. In this context, it should be useful to apply Tukey post hoc, in order to assess prospective significant differences among the different concentrations of P. dactylifera seed extract, and not only vs control group.
- Panels and graphs belonging to the same single figure should be re-organized in size, and arranged in such a way that the entire figure occupies a maximum space of a single page.
- Figure 2: it is not clear to me how intracelular ROS content was evaluated. The experimental group should include evaluation of ROS content by DCF fluorescence intensity in the following experimental groups: Control cells (vehicle-treated only); Control cells + H2O2; Cells treated with P. Dactylifera seed extract at various doses; Cells co-treated with H2O2 and P. Dactylifera seed extract at different doses; Trolox as positive control for antioxidant capacity.Furthermore, the graph should be represented as DCF intensity.
- Even though MTT assay is generally employed as a method to evaluate cell viability, it is actually based on MTT reduction to formazan by several intracellular sources, mainly mitochondria. Thus, MTT assay results should be carefully interpreted, since such variations could reflect alterations in cell metabolism instead of cell viability. I suggest to perform other methods to evaluate cell viability, such as caspase-3 activation, TUNEL assay, or cell count with trypan blue.
- Line 338: please correct "Figure 3A" with "Figure 4A"
- Paragraph 3.6. is entitled "The gene expression levels of tyrosinase, TRP1, TRP2 and MITF were decreased by the P. dactylifera L.
392 seed extract". In my opinion, this title should be changed, according to a different interpretation of the results. Indeed, despite some trends are observable among the experimental groups, no statistical significance is reported, thus I desume that all the experimental groups should be considered identical as the null hypothesis is not rejected - Discussion section, line 506 : the sentence "Our results indicated that P. dactylifera L. seed extract inhibited melanogenesis in B16F10 cells by downregulation of both MAPK and PKA signaling pathways" should be changed, since data about signaling pathways are of correlative nature only.
Author Response
Dear reviewers of Antioxidants
Thank you for reviewing our manuscript entitled “Phoenix dactylifera L. Seed Extract Exerts Antioxidant Properties and Attenuates Melanogenesis in B16F10 Murine Melanoma Cells through Down-Regulation of the MAPK and PKA Signaling Pathways”. Our revisions in response to the reviewers’ comments are addressed below in a point-by-point manner accordingly. Many thanks again for your valuable comments and suggestions. We are looking forward to your positive decision on our article.

Reviewer 2 Report
Huey-Chun and colleagues reported that Phoenix dactylifera L. Seed Extract inhibits melanogenesis coupled with antioxidant activity.
This study is unique, but no descriptions about positive control were described in the manuscript. What is superior compare to present materials for skin whitening?, I cannot understand. Authors should discuss this point more.
How % does ferulic acid contributes to anti-melanogenic activity in the extracts?
It is better to examine ion cheating activity using different ions.
Fig. 4a Why doss Arbutin inhibit tyrosinase so less effective?
Fig. 5 Why doss Arbutin downregulate Mc1R and MITF? Are there any similar reports? Please cite the reference.
minor
Please unify abbreviations, such as TRP-1 or TRP1. (Line79 and material and methods)
FeCl2 “2” is lower. H2O2 also.
Author Response

(The authors gave the same response as above.)

Round 2
Reviewer 1 Report
The authors did not properly addressed most of the reviewer's concerns, and the manuscript still appears rough and superficial, lacking of scientific rigor.
- The authors assessed that they re-organized the panels so that the entire figure occupies a maximum of one page. However, in the new version of the manuscript, I still observe that figure panels are excessively disproportionated. For instance (but not limited to it), Figure 2 is composed by 6 panels which occupies 5 pages. This organization severely impacts the readability of the manuscript. The authors are strongly encouraged to look at the general representation of multipaneled figures in literature (a great number of examples are also obviously available in the most part of the articles published in "Antioxidants").
- The authors performed post-hoc test but did not mention the putative statistical significance among columns. For instance, figure 2, panel B (but not limited to it): are there statistical differences among the three doses of P. dactylifera seed extract? This could be useful to better identify the minimal dose able to induce the most significant biological effect.
- Figure 2F is composed by two panels concerning ROS scavenging capacity (% of control; and % of control NO H2O2). The two panels are identical, except for the statistical analysis. This is very inconsistent.
- Concerning the title of the paragraph 3.6, the change performed by the authors is totally irrelevant. Notably, figure 6 does not show any statistically significant difference, thus it must be assumed that no real differences are present among the diverse experimental groups. The new title “The gene expression levels of tyrosinase, TRP1, TRP2, and MITF were affected by P. dactylifera L. seed extract" still retains the same meaning of the previous title, which is not coherent with the correct interpretation of the data. In other words, the title indicates that P. dactylifera is able to regulate the expression levels of tyrosinase, TRP1, TRP2 and MITF. However, no statistical significance is reported. Thus, it can be assumed that P. dactylifera does not induce any biologically relevant effect. Otherwise, the correct title should be "P. dactylifera seed extract does not affect TRP1, TRP2, and MITF gene expression"
- Concerning the discussion section, the sentence "Our results indicated that P. dactylifera L. seed extract inhibited melanogenesis in B16F10 cells by downregulation of both MAPK and PKA signaling pathways" was changed with "Our results indicated that P. dactylifera L. seed extract inhibited melanogenesis in B16F10 cells by downregulating PKA signaling pathway". This substitution still retains the original meaning that is not correct when considering the obtained results. The data about signaling pathways collected in this work are only correlative, and no causation can be considered. Indeed, no experiments with MAPK and/or PKA inhibitors are performed, thus it is not possible to be sure that changes in p-ERK and p-CREB may be effectively responsible for the effects induced by P. dactylifera on melanogenesis. Again, I suggest the authors to be more restrained in their interpretations, avoiding to draw conclusions that are not supported by their data.
- the authors evaluated CREB phosphorylation (I desume that the phosphorylated residue is Ser133). However, CREB can be activated not only by PKA but also by other kinases such as AKT and Ca2+-calmodulin dependent kinases. Thus, the authors should avoid to mention the involvement of PKA axis, since they did not properly evaluate the activation of PKA. In other words, CREB phosphorylation cannot be considered a univocal readout for PKA activation. From this consideration, the sentence "In this study, P. dactylifera L. seed extract suppressed CREB phosphorylation and PKA activity" is completely misleading, since the authors did not directly evaluate PKA activity.
- The authors should carefully revise the whole manuscript, in order to eliminate the overstatements, and exaggerated interpretations about their results, and this process should be done with caution and accuracy which, in my opinion, are lacking in the present revised version.
Author Response
Dear reviewer:
Many thanks for your valuable comments. We have revised our manuscript carefully as you recommended.

Reviewer 2 Report
The authors reperformed experiments properly.
Author Response
Dear reviewer:
Many thanks for your positive feedback to our revised manuscript.
Round 3
Reviewer 1 Report
Tha authors properly addressed the reviewers' concerns and provided an exhaustive point-by-point response letter.